# Newborn Screening for Spinal Muscular Atrophy: A 2.5-Year Experience in Hyogo Prefecture, Japan

**DOI:** 10.3390/genes14122211

**Published:** 2023-12-14

**Authors:** Shoko Sonehara, Ryosuke Bo, Yoshinori Nambu, Kiiko Iketani, Tomoko Lee, Hideki Shimomura, Masaaki Ueda, Yasuhiro Takeshima, Kazumoto Iijima, Kandai Nozu, Hisahide Nishio, Hiroyuki Awano

**Affiliations:** 1Department of Pediatrics, Kobe University Graduate School of Medicine, 7-5-1, Kusunoki-cho, Chuo-ku, Kobe 650-0017, Japan; sonesho@med.kobe-u.ac.jp (S.S.); ryobo@med.kobe-u.ac.jp (R.B.); ynambu@med.kobe-u.ac.jp (Y.N.); kiiko@med.kobe-u.ac.jp (K.I.); nozu@med.kobe-u.ac.jp (K.N.); 2Hyogo Prefectural Kobe Children’s Hospital, 1-6-7 Minatozimaminami-cho, Chuo-ku, Kobe 650-0047, Japan; iijima@med.kobe-u.ac.jp; 3Department of Pediatrics, Hyogo Medical University, 1-1 Mukogawa-cho, Nishinomiya 663-8501, Japan; to-ri@hyo-med.ac.jp (T.L.); simo-ped@hyo-med.ac.jp (H.S.); ytake@hyo-med.ac.jp (Y.T.); 4Department of Pediatrics, Toyooka Public Hospital, 1094 Tobera, Toyooka 668-8501, Japan; masaaki-ueda@toyookahp-kumiai.or.jp; 5Faculty of Rehabilitation, Kobe Gakuin University, 518 Arise, Ikawadani-cho, Nishi-ku, Kobe 651-2180, Japan; nishio@reha.kobegakuin.ac.jp; 6Research Initiative Center, Organization for Research Initiative and Promotion, Tottori University, 86 Nishi-cho, Yonago 683-8503, Japan

**Keywords:** spinal muscular atrophy, newborn screening, *SMN2* copy number, hybrid gene

## Abstract

Newborn screening (NBS) for spinal muscular atrophy (SMA) is necessary, as favorable outcomes can be achieved by treatment with disease-modifying drugs in early infancy. Although SMA-NBS has been initiated in Japan, its clinical results have not been fully reported. We report the findings of the initial 2.5 years of a pilot SMA-NBS of approximately 16,000 infants conducted from February 2021 in Hyogo Prefecture, Japan. Clinical data of 17 infants who tested positive were retrospectively obtained from the NBS follow-up centers participating in this multicenter cohort observational study. Genetic testing revealed 14 false positives, and three infants were diagnosed with SMA. Case 1 had two copies of survival motor neuron (*SMN*) 2 and showed SMA-related symptoms at diagnosis. Case 2 was asymptomatic, with two copies of *SMN2*. Asymptomatic case 3 had four copies of *SMN2* exon 7, including the *SMN1/2* hybrid gene. Cases 1 and 2 were treated within 1 month and case 3 at 8 months. All the patients showed improved motor function scores and did not require respiratory support. The identification of infants with SMA via NBS and early treatment improved their motor and respiratory outcomes. Thus, implementation of SMA-NBS at a nationwide scale should be considered.

## 1. Introduction

Spinal muscular atrophy (SMA) is a lower motor neuron disease with an autosomal recessive inheritance. Patients exhibit progressive muscular weakness and muscular atrophy [1]. The clinical forms of SMA are classified into five types, from type 0 to type IV, according to the timing of onset and motor developmental achievement. Type I is the most common and severe form, wherein most of the patients become bed-ridden and require ventilator management and tube feeding. Type III and the adult-onset type IV are milder forms, wherein patients do not lose mobility.

The disease-causing gene for SMA is survival motor neuron (*SMN*) 1. Approximately 95% of patients have homozygous deletion of *SMN1,* and 5% of patients retain *SMN1* with an intragenic mutation. The *SMN2* gene, a paralog of *SMN1*, was recognized to be the disease-modifying gene for SMA, since an inverse correlation was observed between the *SMN2* copy number and phenotype [1,2]. Hence, *SMN2* copy number is used as a predictor of disease severity [3].

SMA was recognized as an incurable disease. However, as a result of various clinical trials, three disease-modifying drugs have been clinically available since the late 2010s [1]. Nusinersen, the first Food and Drug Administration–approved drug for the treatment of SMA, was launched in the US in 2016. Furthermore, two other drugs, onasemnogene abeparvovec and risdiplam, are available for clinical use. In Japan, nusinersen was approved by the regulatory authorities in 2017. Subsequently, onasemnogene abeparvovec and risdiplam were approved by the regulatory authorities in 2020 and 2021, and these three drugs are now available in the US and Europe. Nusinersen is an intrathecally administered antisense oligonucleotide that targets *SMN2* pre-mRNA and increases the production of full-length SMN proteins [4]. Onasemnogene abeparvovec is an intravenously administered single-dose gene replacement therapy that carries *SMN* cDNA into an adeno-associated viral serotype 9 vector [5]. Risdiplam is an orally administered small-molecule compound that modifies *SMN2* pre-mRNA splicing and increases the production of full-length SMN proteins [6]. All three drugs have been demonstrated to improve survival and motor outcomes [1,4,5,6]; additionally, early treatment, especially at the presymptomatic stage, maximizes the efficacy of each drug [7,8]. Newborn screening (NBS) for SMA, which detects the homozygous deletion of exon 7 of *SMN1*, has been initiated worldwide [9]. After the introduction of SMA-NBS, the functional burden and associated complications of patients with SMA have improved or are about to improve [10].

The NBS program in Japan covers approximately 20 different diseases. These include endocrine disorders including congenital hypothyroidism and congenital adrenal cortical hyperplasia, and inborn errors of metabolism including phenylketonuria, methylmalonic acidemia, and very long-chain acyl-CoA dehydrogenase deficiency, but not yet SMA. In response to requests from patient and family groups to add SMA to the screening list, the Japanese government has applied for a supplementary budget for research to establish a screening program in 2023. Apart from Japanese government activities, the first prefecture-based study of NBS for SMA began in May 2020 [11]. Since then, NBS has been initiated in several regions, and two infants with SMA were identified via NBS and have received early treatment [12,13].

In this article, we describe the 2.5-year experience of the SMA-NBS pilot study (initiated in February 2021 and one of the earliest in Japan) in Hyogo Prefecture. The outcomes of infants diagnosed with infantile-onset SMA based on clinical symptoms during the treatable era in Japan were compared with the outcomes of patients identified through NBS to demonstrate the efficacy of SMA-NBS.

## 2. Materials and Methods

### 2.1. Study Design

SMA-NBS in Hyogo Prefecture was initiated as a multicenter cohort observational study. The total study period is from 1 February 2021 to 31 March 2031; however, for this study, we analyzed the data up to 31 August 2023. The screening methods for SMA and diagnostic pathways in NBS-positive infants have been reported by Noguchi et al. [11]. Briefly, dried filter paper blood samples were collected from infants at 4–6 days after birth. *SMN1*-detection screening was performed by a real-time quantitative polymerase chain reaction (PCR) assay with hybridization probes, which detects the presence or absence of *SMN1* exon 7, using NeoSMAAT^®^ T/K/S (Sekisui Medical Co., Ltd., Tokyo, Japan). The screening test cost 10,450 yen, and this included the cost of screening for other congenital diseases. Once a positive screening result was identified, the infants were referred to one of the three core facilities in Hyogo Prefecture. Definitive diagnosis was made based on multiple ligation-dependent probe amplification (MLPA) using SALXA MLPA Probemix 021 SMA (MLC-Holland, Amsterdam, The Netherlands), or quantitative real-time PCR as previously reported [14].

### 2.2. Ethics

This study was conducted in accordance with the guidelines of the Declaration of Helsinki and approved by the Ethics Committee of Kobe University Graduate School of Medicine (No. B200086; approved on 24 June 2020). Written informed consent for publication was obtained from the parents of all the patients.

### 2.3. Clinical Data

Patient data were retrospectively obtained from the electronic medical records of NBS follow-up centers. To characterize the clinical courses of patients with SMA detected with NBS, the clinical data of patients based on clinical symptoms were obtained. Patients with SMA diagnosed based on clinical symptoms were recruited if they met all of the following criteria: (1) born after August 2017, when disease-modifying drugs became clinically available in Japan; (2) diagnosed within one year of birth (infantile onset); (3) followed up at the Department of Pediatrics, Kobe University Hospital.

From data of height and body weight, standard deviation (SD) scores were calculated using the physique index calculation software (ver3.3) built by the Japanese Society for Pediatric Endocrinology (http://jspe.umin.jp/medical/taikaku.html, accessed on 23 October 2023).

## 3. Results

### 3.1. Demographic and Genetic Characteristics of the Newborns Identified as Positive during the SMA-NBS Pilot Study

Between 1 February 2021 and 31 August 2023, 26 medical institutions from Hyogo Prefecture participated in the study. A total of 16,037 infants were included in the study. Among the investigated infants, 17 (0.11%) tested positive for SMA-NBS (Figure 1). The positive results were reported by the laboratory at a median age of 20 days (range: 13–24 days). The positive infants were examined by the NBS follow-up center at a median age of 23 days (14–35 days). All the positive infants were confirmed via genetic testing at a median age of 24 days (19–41 days). Three infants with SMA were found to exhibit a homozygous deletion of *SMN1* exon 7, resulting in a birth prevalence of 1:5346. The remaining 14 infants had two copies of *SMN1* exon 7, suggesting false positives. Among those who underwent SMA-NBS in Hyogo Prefecture, no case was diagnosed with SMA based on clinical symptoms.

### 3.2. Issues with Sample Preparation for SMA-NBS

Fourteen infants identified as false positives were born at five different medical institutes. All the institutes were core hospitals with neonatal intensive care units. Many infants identified as false positives were born at institutes that used heparinized blood to prepare samples [11]. No infants identified as false positives have been reported since December 2022, after the request to suspend the use of heparinized blood for sample preparation. The number of infants who were born in five core hospitals before and after December 2022 was 3010 and 1711, respectively. The false positive rate before December 2022 was 0.93%.

### 3.3. Clinical Couser in the Patients with SMA Identified via NBS

#### 3.3.1. Case 1 (Patient ID SZ17)

The patient, a 17-month-old female infant, was symptomatic at the time of diagnosis at the age of 23 days (Table 1). She had two copies of *SMN2* detected by genetic testing using MLPA (Figure 2). Clinical courses for up to six months of age have been reported [11]. After four doses according to the nusinersen loading protocol, onasemnogene abeparvovec treatment was administered at the age of 4 months (Table 2). The patient gained neck control and balance at 7 months of age and rolled over at 8 months. The patient was able to walk with support at 15 months. The Children’s Hospital of Philadelphia Infant Test of Neuromuscular Disorders (CHOP-INTEND) scores increased to 44, 55, and 58 at 5, 8, and 12 months of age, respectively (Figure 3). The patient had advanced scoliosis and was trained to sit with braces. Due to poor weight gain (5.5 kg, −1.8 SD), tube feeding was initiated at the age of 5 months; the patient is currently able to swallow pasteurized food. Her height and weight (SD) at 15 months are 74.5 cm (−0.77 SD) and 8.8 kg (−0.57 SD), respectively. The patient did not require respirators. She had scoliosis with a Cobb angle of 37°. The compound muscle action potential (CMAP) of the median nerve increased from a baseline value of 0.08 mV [11] to 0.97 mV.

#### 3.3.2. Case 2 (Patient ID SZ18)

The patient, a 14-month-old female infant, had two copies of *SMN2* and was asymptomatic at the time of diagnosis on day 19 (Figure 2) (Table 1). Clinical courses up to 3 months of age have been reported [11]. After five doses according to the nusinersen administration protocol, onasemnogene abeparvovec treatment was administered at the age of 8 months after obtaining family consent. Although the patient presented with mild weakness and proximal muscle predominance in the lower extremities and X legs, she acquired normal gross motor milestones (Table 2). Her CHOP-INTEND score increased, reaching a perfect value of 64 at 6 months (Figure 3). Her height and weight at the age of 12 months were 71.0 cm (−1.36 SD) and 8.2 kg (−0.85 SD), respectively. She did not require tube feeding or respiratory support and had no orthopedic problems. The ulnar nerve CMAP has been within the normal range and increased from 4.4 mV at 7 months to 7.5 mV at 16 months.

#### 3.3.3. Case 3 (Patient ID Z19)

The patient, an 11-month-old male infant, is the third child of nonconsanguineous parents. None of the family members had been previously diagnosed with a neuromuscular disorder. He was born via scheduled cesarean section at 38 weeks and 2 days of gestation, weighing 3009 g. After birth, the patient showed no symptoms related to SMA. At the age of 24 days, he tested positive for SMA-NBS and was referred to a follow-up institution on the same day (Table 1). MLPA revealed 0 and 1 copies of *SMN1* exons 7 and 8, respectively (Figure 2) and confirmed the diagnosis of SMA on day 24. MLPA also revealed 4 copies of *SMN2* exon 7 and 3 copies of *SMN2* exon 8, indicating that the patient had one copy of the hybrid *SMN* gene, in addition to three copies of normal *SMN2*. As for a hybrid gene, when the copy numbers of *SMN2* exon 7 are the same in the *SMN1*-deleted patients, there are no obvious differences between those with and those without a hybrid gene (*SMN2* exon 7 -*SMN1* exon 8). Based on our experience, we considered the patient just as we would a case of a *SMN1*-deleted patient with four copies of *SMN2*. The clinical severity of our previous patients with a hybrid gene has already been reported elsewhere [16]. At the time of diagnosis, the patient had no symptoms related to SMA. The CMAP of the ulnar nerve at the age of 30 days was within the normal range of 3.4 mV.

During follow-up, the patient’s 2-year-old sibling presented with a motor developmental delay. She acquired independent walking at the age of 15 months but gradually became prone to stumbling. She exhibited an X leg and decreased deep tendon reflexes. MLPA revealed that the sibling had the same genotype as the present case, confirming the diagnosis of SMA type III. The sibling initiated nusinersen treatment at the age of 2 years and 7 months.

After his sibling’s diagnosis and the treatment decision-making session, onasemnogene abeparvovec was administrated at the presymptomatic stage (Table 2). Currently, the patient is showing normal motor development without any swallowing or respiratory problems.

### 3.4. Outcomes in the Patients with SMA Diagnosed Based on Clinical Symptoms

#### 3.4.1. Case 4 (Patient ID SZ15)

The patient, a 2-years-and-3-months-old girl, presented with hypotonia, which commenced at approximately 2 weeks of age (Table 1). As her hypotonia became marked, she visited a hospital at the age of 2 months and was referred to a referral hospital at 3 months of age. MLPA revealed a homozygous deletion of *SMN1* and two copies of *SMN2*; accordingly, the patient was diagnosed with SMA type I (Figure 1). The first nusinersen treatment was administered at the age of 103 days in a total of five doses, followed by onasemnogene abeparvovec at the age of 7 months. The patient’s CHOP-INTEND score was 47 at 15 months (Figure 3). At the age of 5 months, her respiratory condition worsened due to an upper respiratory tract infection. She was intubated and subsequently underwent tracheostomy due to difficulty in extubation (Table 1). Furthermore, she underwent gastrostomy at 12 months of age. Her median CMAP was 0.22 mV at the age of 16 months.

#### 3.4.2. Case 5 (Patient ID ZS16)

A 3-years-and-3-months-old boy had visited a referral hospital on day 14 after birth because of poor feeding and was diagnosed with SMA type I due to homozygous deletion of *SMN1* and two copies of *SMN2* (Table 1) (Figure 2). His sibling exhibited type I SMA. The patient received onasemnogene abeparvovec treatment at 50 days [15]. Subsequently, owing to a decrease in the CHOP-INTEND score (Figure 3), nusinersen treatment was initiated at the age of 14 months and is still ongoing. The CHO-INTEND score at thelatest visit was 40. The patient did not acquire head control and required respiratory ventilation and full feeding via gastrostomy (Table 2). CMAP has not been measured in this patient.

## 4. Discussion

### 4.1. Patients with Two SMN2 Copies

In the three patients with SMA identified via SMA-NBS in this study, two patients had two *SMN2* copies. Govoni et al. reported that in SMA type 1, the therapeutic window can be hypothesized to be, optimally, within 1 month [17]. Our patients with two *SMN2* copies initiated early treatment at the best possible time. In Germany, fifteen patients with two *SMN2* copies identified with NBS have been reported [18]. They showed increased motor function scores with early nusinersen therapy within days 14–39, with or without pretreatment symptoms. None of the patients had orthopedic complications or required ventilatory management or tube feeding at the last visit (7 weeks–26 months) [18]. When treated early, patients with two copies do not always exhibit a normal neurological course. For instance, Schwartz et al. reported 21 patients with two *SMN2* copies treated within 6 weeks of birth, of which 6 (29%) showed proximal predominant muscle weakness and 3 showed clinical signs of SMA type II–like symptoms at 16–30 months [19]. Schwartz et al. also reported that the CHOP-INTEND score at baseline may be a prognostic indicator of motor outcomes. A score of >50 suggests a good prognosis, a score of <30 indicates a less favorable prognosis, and a score of 30–50 is not associated with prognosis [19]. Our results support these findings. Case 1, with a score of 12, exhibited SMA type II-like symptoms. However, after treatment, Case 1 acquired a score of 42 and showed mild proximal predominant muscle weakness but generally acquired normal gross motor function.

In the patients with two *SMN2* copies in this study, one required tube feeding. Predicting the need for tube feeding before treatment is still difficult. Crawford et al. reported the prognosis for swallowing problems in patients with SMA who received nusinersen at the presymptomatic stage. A total of 5 out of 25 patients required tube feeding during a median observation period of 4.9 years [20]. Of these 5 patients, 1 had low baseline peroneal CMAP amplitudes (<2 mV) and 3 had low ulnar CMAP amplitudes (<2 mV). Patients with low CMAP may exhibit progressed peripheral neuropathy and may be less responsive to treatment. Our patient, who received nusinersen therapy at the presymptomatic stage, showed an ulnar CMAP of 3.5 mV at baseline and required no feeding support. The clinical course of patients diagnosed with NBS and treated early differs from that of patients without disease-modifying treatment. However, even in cases of SMA detected with NBS, early treatment may not always be a “cure”, depending on the condition at the time of diagnosis. Motor function scores and CMAP are important indicators for appropriate intervention in terms of the prevention of complications and subsequent rehabilitation after treatment.

### 4.2. SMA Patient with an SMN1/SMN2 Hybrid Gene

Accurate and rapid determination of the *SMN2* copy number in patients identified through SMA-NBS is critical for determining the timing of treatment and the therapeutic agents [21,22]. Some patients have a hybrid gene containing exon 7 of *SMN2* and exon 8 of *SMN1*. Patients with such hybrid genes have been reported in various ethnic groups, including the Japanese population [15,23,24,25,26,27]. We present the first case of an SMA patient with a hybrid gene identified with NBS. No stated recommendations are available for the treatment of hybrid cases, and the indications for treatment have not yet been established [28]. In Case 3, the MLPA results indicated that in addition to three copies of normal *SMN2*, one copy of the hybrid gene was present. This hybrid gene was presumably produced by gene conversion from *SMN1* to *SMN2* [1]. Since the copy number of exon 7 of *SMN2* correlates with the clinical presentation of patients with SMA [15,28,29], Case 3, with four copies of exon 7 of *SMN2,* was considered as having SMA type III. In fact, the older sibling, with an identical genotype as in Case 3, had SMA type III. At that time, no treatment protocol had been established for 4-copy cases identified through SMA-NBS in Japan. After informing his parents of the diagnosis, multiple treatment decision-making sessions were conducted. A careful monitoring of the clinical symptoms was decided on the basis of the recommendation proposed by Glascock et al. [22]. The younger sibling (Case 3) was asymptomatic and achieved normal gross motor milestones until 8 months. After the older sibling’s diagnosis of SMA type III, we conducted treatment decision-making session again. Since the patient, as well as his sibling, was at risk of developing the disease before the age of 2 years, a pre-symptomatic treatment was decided upon. Compared to his older sibling, he received earlier treatment thanks to SMA-NBS. We hope that long-term follow-up data from this patient will provide evidence to examine the benefits of early treatment in hybrid gene cases detected using NBS.

### 4.3. Treatment Decision-Making Support for Families of Patients with SMA Identified via NBS

Early diagnosis following positive SMA-NBS results encourages the patient’s family to accept the disease and make treatment-related decisions in a short period of time. Diagnosis through NBS generates feelings of shock, fear, anxiety, disbelief, or a combination of these in parents [30]. Additionally, engaging with numerous and diverse medical teams can be overwhelming [31]. The mean age at the initiation of treatment of patients with two copies of *SMN2* detected with NBS and a high need for early treatment was 23 days (11–89 days) [32]. The two-copy patients in this study also received their first treatment within the first month (22 and 25 days of age). Timely notification and education are crucial, considering the emotional and psychological states of parents. Shared treatment decision making with parents is necessary [33]. In our experience, parents cannot always accept the disease, fully understand the characteristics of multiple treatments, or initiate treatment without delay. In our NBS follow-up facilities, experienced pediatric neurologists as well as clinical geneticists, nurses, genetic counselors, psychologists, and medical social workers are involved in the diagnosis, treatment, and subsequent follow-up to assist the parents in decision making. With the help of this team, the parents of our two-copy patient complied with the treatment protocol of nusinersen and subsequent abeparvovec onasemnogene therapy.

### 4.4. Cost-Effectiveness of SMA-NBS

SMA-NBS has already been implemented in many countries, and not only the clinical usefulness of NBS but also its cost-effectiveness has been examined. Weidlich et al. examined the cost effectiveness of SMA-NBS in the UK [34]. They estimated NBS identified in approximately 56 infants with SMA per year, 46 presymptomatic and 10 symptomatic. After comparing scenarios with and without NBS, they concluded that NBS improves the health outcomes of SMA patients and is less costly. In the Netherlands, Velikanova et al. compared health outcomes and costs of infants with SMA identified via NBS and in those diagnosed after the onset of the disease [35]. Similar to the UK results, NBS improved health outcomes and is less costly. In Australia, Shih et al. reported that compared with no NBS and late nusinersen treatment, infants with SMA who were identified by NBS and received early gene therapy had improved quality of life and length of life [36]. They concluded that SMA-MBS would be value-for-money from an Australian clinical and policy context. These results indicate that SMA-NBS is a cost-effective use of resources. Unfortunately, there have been no studies on the cost-effectiveness of SMA-NBS in Japan. Since our study has demonstrated the clinical usefulness of NBS, it is hoped that an economic study will be conducted in Japan in the future.

### 4.5. Challenges in Implementing SMA-NBS in Japan

Implementing nationwide SMA-NBS in Japan is challenging due to low awareness regarding SMA. In a survey of the general population in Japan, the awareness of SMA was approximately 50%, and >90% of people were unaware of the treatable nature of SMA [37]. However, >95% supported SMA-NBS, indicating a high potential need. Increased awareness amongst the general population is needed in the future. SMA-NBS may reveal the status of carriers in newborns and their parents. The possibility that asymptomatic carriers or their relatives may be differentiated based on their genetic characteristics should be considered. In a survey conducted among the general Japanese population, approximately 3% of the respondents reported encountering undesirable experiences when seeking genetic information related to marriage, pregnancy, and childbirth [38]. In Japan, a new law on the promotion of genomic medicine and the prevention of genetic discrimination has just been passed in 2023. We expect that increased awareness regarding the human genome and respect for its diversity can lead to increased acceptance of SMA-NBS.

### 4.6. Limitations

In this study, we collected the data retrospectively. This pilot study is currently underway. Therefore, the number of newborns undergoing SMA-NBS was small and the study duration was too short to estimate the additional long-term benefits of SMA-NBS. Since follow-up of SMA-NBS–negative infants has not been conducted, accurate data on false negatives are not available.

## 5. Conclusions

In this paper, we present the results of the SMA-NBS pilot study conducted in Hyogo Prefecture, Japan. During the 2.5-year period, we identified three patients with SMA using NBS. Two of the patients had two copies of *SMN2* and one had four copies of *SMA2*; of the patients with two copies, one was before and one after the disease onset.

There were short intervals between referral to the NBS follow-up center, diagnosis, and subsequent treatment. Compared to patients with two copies of *SMN2* identified based on clinical symptoms, patients identified by NBS were diagnosed and treated earlier and had better outcomes.

Timely diagnosis and treatment can improve motor and respiratory functions in patients with SMA. Thus, implementation of SMA-NBS at a nationwide scale should be considered. Additionally, further large-scale prospective studies are required to assess the long-term outcomes of SMA-NBS.

## Figures and Tables

**Figure 1 genes-14-02211-f001:**
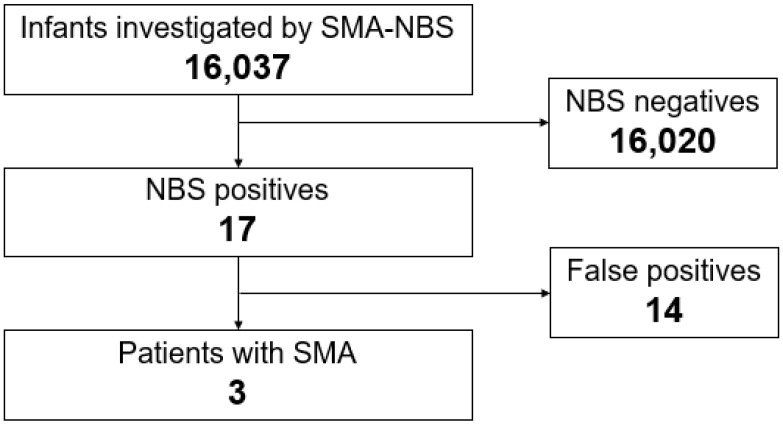
Flowchart of the newborn screening for SMA.

**Figure 2 genes-14-02211-f002:**
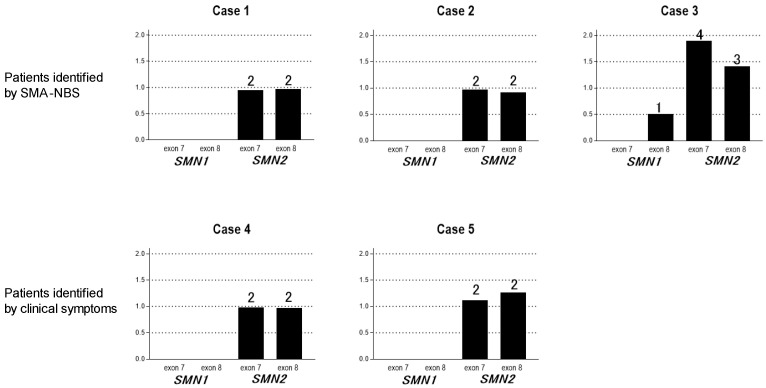
Multiple ligation probe amplification (MLPA) or quantitative real-time polymerase chain reaction (PCR) results of *SMN1* and *SMN2.* Black bars indicate relative peak ratio of the MLPA products of the *SMN1* and *SMN2* exons in cases 1–4. Black bar in case 5 indicates quantitative real-time PCR product. The number above the bar indicates the number of copies of *SMN2.* Case 3 had four copies of *SMN2* exon 7 and three copies of *SMN2* exon 8, indicating that the patient had one copy of the hybrid *SMN* gene. Refer to the main text for details.

**Figure 3 genes-14-02211-f003:**
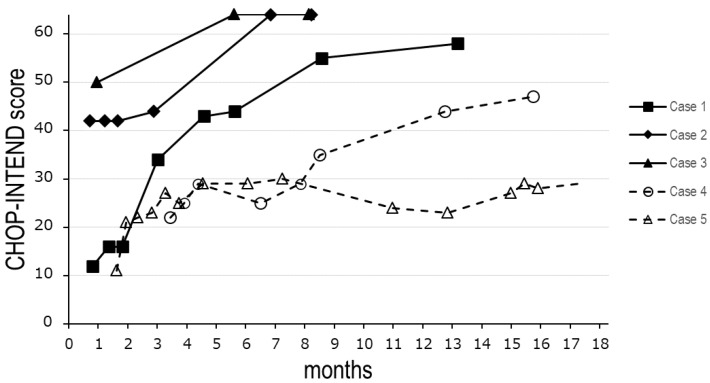
Changes in the CHOP-INTEND score after treatment. Solid symbols and solid lines indicate the changes in the CHOP-INTEND score in cases identified via SMA-NBS. Open symbols and dashed lines indicate the changes in the score in cases detected based on clinical symptoms.

**Table 1 genes-14-02211-t001:** Summary of the characteristics of the patients.

Case	1 *	2 *	3	4	5 **
ID	SZ17	SZ18	Z19	SZ15	ZS16
Sex	F	F	M	F	M
Current age	17 months	13 months	11 months	2 y 3 m	3 y 3 m
*SMN2* exon 7 copy number	2	2	4	2	2
NBS	+	+	−	−	−
Age at onset	11 days	−	−	2 weeks	After birth
First symptom identified	Hypotonia	−	−	Hypotonia	Feeding related problem
Visit to referral center	21 days	18 days	22 days	95 days	14 days
Diagnosis	24 days	21 days	24 days	96 days	28 days

NBS: newborn screening. * Initial clinical course reported [11]. ** Clinical course up to 5 months reported [15]. “+” indicates “received”. “−” indicates “not received” or “not observed”.

**Table 2 genes-14-02211-t002:** Patient outcomes following treatment.

Case	1 *	2 *	3	4	5 **
Age at initiation of SMA treatment (drug)	25 days (Nusinersen)	22 days (Nusinersen)	260 days (Onasemnogene abeparvovec)	103 days (Nusinersen)	50 days (Onasemnogene abeparvovec)
Age at initiation of combination treatment	4 months (Onasemnogene abeparvovec)	8 months (Onasemnogene abeparvovec)	−	7 months (Onasemnogene abeparvovec)	15 months (Nusinersen)
Age at head control	7 months	4 months	4 months	15 months	Not achieved
Age at sitting with support	9 months	7 months	7 months	Not achieved	Not achieved
Age at sitting without support	Not achieved	8 months	7 months	Not achieved	Not achieved
Age at walking with support	Not achieved	13 months	8 months	Not achieved	Not achieved
Age at independent walking	Not achieved	14 months	8 months	Not achieved	Not achieved
Respiratory support (age)	Spontaneous breathing	Spontaneous breathing	Spontaneous breathing	NPPV (3 months) Tracheostomy (5 months)	NPPV (14 months) Tracheostomy (24 months)
Nutritional support (age)	Nasogastric tube (5 months)	Oral feeding	Oral feeding	Gastrostomy tube (12 months)	Nasogastric tube (4 months) Gastrostomy tube (23 months)
Orthopedic complications	Scoliosis	−	−	Scoliosis	Scoliosis Thoracic deformity

NPPV: noninvasive positive pressure ventilation, SMA: spinal muscular atrophy. * Initial clinical course reported [11]. ** Clinical course up to 5 months reported [15]. “−” indicates “not received” or “not observed”.

## Data Availability

The data supporting the findings of this study are available from the corresponding author H.A. upon reasonable request.

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
