# Peer review of "Newborn Screening for Spinal Muscular Atrophy: A 2.5-Year Experience in Hyogo Prefecture, Japan"

_genes, 2023, doi:10.3390/genes14122211_

Round 1

Reviewer 1 Report

Comments and Suggestions for Authors

Dear author(s),

Thank you for your interesting real-world experience regarding new-born screening for SMA. This is a very interesting and clinically important topic.

I have several minor suggestions:

-Within the introduction section I would add a paragraph commenting on new-born screening programmes in Japan in general and what do they test for. Also would comment if SMA is included or if government and healthcare recently speculated about inclusion. This would make readers more familiar with the setting you are writing about.

-Regarding methodology - I would add some more data on type of test used for SMA, its availability, cost, procedure description, etc.

-Also, several recent articles are talking about the cost-effectiveness of new-born screening programme (taking into account early DMD introduction and better treatment effectiveness). I would add that to improve the discussion.

Best regards, Peer-reviewer

Author Response

Thank you for your valuable comments. I responded to the comments in a question-and-answer format. Corrections or additions to the main text are highlighted in yellow.

Dear author(s),

Thank you for your interesting real-world experience regarding new-born screening for SMA. This is a very interesting and clinically important topic. I have several minor suggestions:

-Within the introduction section I would add a paragraph commenting on new-born screening programmes in Japan in general and what do they test for. Also would comment if SMA is included or if government and healthcare recently speculated about inclusion. This would make readers more familiar with the setting you are writing about.

Thank you for valuable comment. I have added a paragraph regarding the current newborn screening program in Japan and future prospect of SMA-NBS by the Government is section 1, as follows:

“The NBS program in Japan covers approximately 20 different diseases. These include endocrine disorders including congenital hypothyroidism and congenital adrenal cortical hyperplasia, and inborn errors of metabolism including phenylketonuria, methylmalonic acidemia, and very long-chain acyl-CoA dehydrogenase deficiency, but not yet SMA. In response to request from patient and family groups to add SMA to the screening list, the Japanese government has applied for a supplementary budget for research to establish a screening program in 2023. Apart from Japanese government activities, “

-Regarding methodology - I would add some more data on type of test used for SMA, its availability, cost, procedure description, etc.

Thank you for your comment. I have added more data in section 2 as follows:

“Briefly, dried filter paper blood samples were collected from the infants at 4-6 days after birth. SMN1-detection screening were performed by a real-time quantitative polymerase chain reaction assay with hybridization probes, which detects the presence or absence of SMN1 exon 7, using NeoSMAAT® T/K/S(Sekisui Medical Co., Ltd., Tokyo, Japan ). The screening test costs 10,450 yen, but this includes the cost of screening for other congenital diseases. Once a positive screening result is identified, the infants is referred to one of the three core facilities in Hyogo Prefecture. Definitive diagnosis was made based on multiple ligation-dependent probe amplification (MLPA) using SALXA MLPA Probemix 021 SMA (MLC-Holland, Amsterdam, The Netherlands) or quantitative real-time PCR as previously reportted [14].“

-Also, several recent articles are talking about the cost-effectiveness of new-born screening programme (taking into account early DMD introduction and better treatment effectiveness). I would add that to improve the discussion.

Thank you for your comment to improve our manuscript. I have added discussion about cost-effectiveness of SMA-NBS as follows:

“4.4 Cost-effectiveness of SMA-NBS

SMA-NBS has already been implemented in many countries, and not only the clinical usefulness of NBS but also its cost-effectiveness has been examined. Weidlich et al. examined the cost effectiveness of SMA-NBS in the UK [34]. They estimated NBS identified approximately 56 infants with SMA per year, 46 presymptomatic and 10 symptomatic. After comparing scenarios with and without NBS, they concluded that NBS improves the health outcomes of SMA patients and is less costly. In the Netherlands, Velikanova et al. compared health outcomes and costs of infants with SMA identified via NBS and in those diagnosed after onset of disease.[35]. Similar to the UK results, NBS improved health outcomes and is less costly. In Australia, Shih et al. reported that compared with no NBS and late nusinersen treatment, infants with SMA who were identified by NBS and received early gene therapy had improved quality of life and length of life [36]. They concluded that SMA-MBS would be value-for-money from an Australian clinical and policy context. These results indicate that SMA-NBS is a cost-effective use of resources. Unfortunately, there have been no studies on the cost-effectiveness of SMA-NBS in Japan. Since our study has demonstrated clinical usefulness of NBS, it is hoped that an economic study will be conducted in Japan in the future. “

While reviewing, we noticed that The SMN2 copy number measurement in case 5 was a quantitative real-time PCR method, not MLPA. Therefore, the method (section 2) was modified. The Fig.2 was revised and the result of case 5 was deleted from Fig. 2. The explanation of SMN2 in case 5 was added to section 3.3. I am sorry for the careless mistake.

I also found a spelling error Reference 13 and I has corrected. (Kamuzu to Kimizu)

Reviewer 2 Report

Comments and Suggestions for Authors

Comments and Suggestions for Authors

This manuscript presents the results obtained in a 2,5-year time-span from the SMA-NBS pilot programme in Japan. There are important observations and comments in the Discussion, the overall scientific contribution of the study is important in considering the implementation of SMA-NBS not only in Japan but on international level.

Comments

1.     Introduction-paragraph 3: The newborn screening (NBS) for SMA detects the homozygous deletion of exon7 of SMN1?

2.     In the same paragraph the last sentence is grammatically incorrect

3.     In paragraph 3.3 Case 1 female – he should be corrected to correct gender

4.     For better comparison, in Table 2. the age at initiation of treatment should be given in days for all patients

5.     The first paragraph of the Conclusions section: Sentence “Two of these patients had two copies..“ is confusing: “one before and one after the disease onset”. Should be rephrased.

6.     Based on the data provided in Table 1. and 2. it is difficult to conclude that by NBS treatment was initiated earlier and had better outcomes in general.

Questions

1.     In paragraph 3.2: Could you please give the number/percentage of FP samples from heparinised blood and also the number of samples before and after December 2022. The reference 11 states that “..despite this request, heparinized blood samples were prepared for SMA-NBS after May 2022, causing further false positives (cases 8–10)”

2.     Was there any difference observed in SMA patients found based on symptoms or NBS with a hybrid gene compared to those with SMN2 copies?

Suggestions

It would give a good support for the scientific contribution of the study if more patients would be compared based on clinical data of disease status.

Comments on the Quality of English Language

Typo and editing errors should be corrected

Author Response

Comments and Suggestions for Authors

This manuscript presents the results obtained in a 2,5-year time-span from the SMA-NBS pilot programme in Japan. There are important observations and comments in the Discussion, the overall scientific contribution of the study is important in considering the implementation of SMA-NBS not only in Japan but on international level.

Thank you for your valuable comments. I responded to the comments in a question-and-answer format. Corrections or additions to the main text are highlighted in yellow.

Comments

  1. Introduction-paragraph 3: The newborn screening (NBS) for SMA detects the homozygous deletion of exon7of SMN1?

Thank you for your comment. You are correct. I have added exon 7 to sentence.

  1. In paragraph 3.3 Case 1 female – he should be corrected to correct gender

Thank you for your careful review. I have made the corrections.

  1. For better comparison, in Table 2. the age at initiation of treatment should be given in days for all patients

Thank you for your comment. I have corrected table 2 to indicate the timing of treatment initiation in days old.

  1. The first paragraph of the Conclusions section: Sentence “Two of these patients had two copies..“ is confusing: “one before and one after the disease onset”. Should be rephrased.

Thank you for your comment. I have rewritten the sentence in section 5.

“Two of the patients had two copies of SMN2 and one had four copies of SMA2; of the patients with two copies, one before and one after the disease onset.”

  1. Based on the data provided in Table 1. and 2. it is difficult to conclude that by NBS treatment was initiated earlier and had better outcomes in general.

Thank you for your valuable comment.

Our patients with two SMN2 copies who were expected to be the severe form (SMA type 1) was treated within 1 month after birth. Govoni reported that in SMA type 1, according to the exper- imental data obtained so far, the therapeutic window can be hypothesized to be, optimally, within 1 month of age or even within few weeks after birth (Govoni A, et al. Molecular Neurobiology. 2018;55,6307-18. DOI: 10.1007/s12035-017-0831-9). Therefore, initiation of treatment within one month is important in terms of early treatment. Compared with patients found by clinical symptoms, patients identified by NBS received early treatment at the best possible time. I cited Govoni’s report and added in the text that for the severe form, patients identified with NBS were able to start treatment within one month.

For the patients with four SMN2 copies, ccomparison is difficult because there was only one patient with 4 copies. I added the following sentence in section 3.3 and in the discussion, “The sibling initiated nusinersen treatment at the age of 2 years and 7 months. After his sibling’s diagnosis and the treatment decision-making session, onasemnogene abeparvovec administrated at the presymptomatic stage (Table 2).”, and “Compared to his older sibling, he received earlier treatment thanks to SMA-NBS.”, respectively

As I mentioned in 4.1 discussion part, even in cases of SMA detected with NBS, early treatment may not always be a “cure,“ depending on the condition at the time of diagnosis. Case 1 who was symptomatic at the time of initial treatment acquired sitting with support, but Case 4 and 5 with no NBS and treatment after one month of age did not. I added the item “age at sitting with support” in Table 2. I also added the idem “age at walking with support” to show more detailed data.

There was an error in writing, I corrected the age at head control in Case 2.

Questions

  1. In paragraph 3.2: Could you please give the number/percentage of FP samples from heparinised blood and also the number of samples before and after December 2022. The reference 11 states that “..despite this request, heparinized blood samples were prepared for SMA-NBS after May 2022, causing further false positives (cases 8–10)”

Thank you for your comment.

The number of infants who were born in 5 core hospitals with neonatal intensive care unit before and after December 2022 were 3010 and 1711, respectively. Therefore, the false positive rate at 5 hospitals was 0.93% (14/1513). I am sorry that the number of heparin blood samples is unknown, as the exact data was not collected.

I have added these results to section 3.2

  1. Was there any difference observed in SMA patients found based on symptoms or NBS with a hybrid gene compared to those with SMN2 copies?

Thank you very much for your kind suggestion.   Following your suggestion, we revised manuscript and added the next sentences in the description of 3.3. Case 3,

“As for a hybrid gene, when the copy numbers of SMN2 exon 7 are the same in the SMN1-deleted patients, there are no obvious differences between those with or without a hybrid gene (SMN2 exon 7 -SMN1 exon 8). Based on our experience, we considered the patient just as a case of SMN1-deleted patient with four copies of SMN2. The clinical severity of our previous patients with a hybrid gene has already been reported elsewhere [15]”

Suggestions

It would give a good support for the scientific contribution of the study if more patients would be compared based on clinical data of disease status.

Thank you for your valuable suggestions. We wanted to include as much data as possible from many patients in this study, too. However, since the drug became available for clinical use in Japan, there have been only two patients born in Hyogo Prefecture with SMA who developed the disease within the first year of life. As our study continues, we will continue to collect data on cases of NBS-identified and patients diagnosed on the basis of clinical symptoms without NBS.

~~~~~~~~~~~~~~~~~~~~~~~~~~~

While reviewing, we noticed that The SMN2 copy number measurement in case 5 was a quantitative real-time PCR method, not MLPA. Therefore, the method (section 2) was modified. The Fig.2 was revised and the result of case 5 was deleted from Fig. 2. The explanation of SMN2 in case 5 was added to section 3.3. I am sorry for the careless mistake.

I also found a spelling error Reference 13 and I has corrected. (Kamuzu to Kimizu)
